# RNA Polymerase II Transcription in *Pneumocystis*: TFIIB from *Pneumocystis carinii* Can Replace the Transcriptional Functions of Fission Yeast *Schizosaccharomyces pombe* TFIIB In Vivo and In Vitro

**DOI:** 10.3390/ijms23126865

**Published:** 2022-06-20

**Authors:** Diego A. Rojas, Fabiola Urbina, Aldo Solari, Edio Maldonado

**Affiliations:** 1Instituto de Ciencias Biomédicas (ICB), Facultad de Ciencias de la Salud, Universidad Autónoma de Chile, Santiago 8910132, Chile; 2Programa de Biología Celular y Molecular, ICBM, Facultad de Medicina, Universidad de Chile, Santiago 8380492, Chile; fabiola.urbina1516@gmail.com (F.U.); asolari@uchile.cl (A.S.)

**Keywords:** *Pneumocystis carinii*, transcription, general transcription factors (GTFs), RNA polymerase II, *Schizosaccharomyces pombe*

## Abstract

The *Pneumocystis* genus is an opportunistic fungal pathogen that infects patients with AIDS and immunocompromised individuals. The study of this fungus has been hampered due to the inability to grow it in a (defined media/pure) culture. However, the use of modern molecular techniques and genomic analysis has helped researchers to understand its complex cell biology. The transcriptional process in the *Pneumocystis* genus has not been studied yet, although it is assumed that it has conventional transcriptional machinery. In this work, we have characterized the function of the RNA polymerase II (RNAPII) general transcription factor TFIIB from *Pneumocystis carinii* using the phylogenetically related biological model *Schizosaccharomyces pombe*. The results of this work show that *Pneumocystis carinii* TFIIB is able to replace the essential function of *S. pombe* TFIIB both in in vivo and in vitro assays. The *S. pombe* strain harboring the *P carinii* TFIIB grew slower than the parental wild-type *S. pombe* strain in complete media and in minimal media. The *S. pombe* cells carrying out the *P. carinii* TFIIB are larger than the wild-type cells, indicating that the TFIIB gene replacement confers a phenotype, most likely due to defects in transcription. *P. carinii* TFIIB forms very weak complexes with *S. pombe* TATA-binding protein on a TATA box promoter but it is able to form stable complexes in vitro when *S. pombe* TFIIF/RNAPII are added. *P. carinii* TFIIB can also replace the transcriptional function of *S. pombe* TFIIB in an in vitro assay. The transcription start sites (TSS) of the endogenous *adh* gene do not change when *P. carinii* TFIIB replaces *S. pombe* TFIIB, and neither does the TSS of the *nmt1* gene, although this last gene is poorly transcribed in vivo in the presence of *P. carinii* TFIIB. Since transcription by RNA polymerase II in *Pneumocystis* is poorly understood, the results described in this study are promising and indicate that TFIIB from *P. carinii* can replace the transcriptional functions of *S. pombe* TFIIB, although the cells expressing the *P. carini* TFIIB show an altered phenotype. However, performing studies using a heterologous approach, like this one, could be relevant to understanding the basic molecular processes of *Pneumocystis* such as transcription and replication.

## 1. Introduction

Eukaryotic RNA polymerase II (RNAPII) is a multisubunit enzyme, which transcribes messenger RNAs, microRNAs, small nuclear RNAs and long non-coding RNAs [1,2]. RNAPII is recruited to the gene promoters by a set of general transcription factors (GTFs) in order to form a preinitiation complex (PIC). The set of GTFs includes TFIIA, TFIIB, TFIID (composed by TATA-binding protein: TBP and TBP-associated factors: TAFs), TFIIE, TFIIF and TFIIH, which are defined as the minimal set to in vitro transcribe a gene containing a TATA-box promoter element [1,2]. The GTFs and RNAPII are recruited to the gene promoters in an ordered fashion, since TFIID is the first factor which binds to the TATA-box, followed by TFIIA and TFIIB [1,2]. Afterwards, the complex RNAPII-TFIIF is recruited to the complex followed by TFIIE and TFIIH. Following the assembly of the PIC and in the presence of the ribonucleotides triphosphate, transcription initiation takes place [2].

The GTF TFIIB is a single polypeptide, and plays key roles in PIC formation, since it bridges promoter-bound TFIID with the RNAPII/TFIIF complex [3,4,5]. TFIIB is remarkably conserved through plants, animals, and fungi, since it exhibits structural similarity both at the level of primary sequence and at the structural level [6]. It is involved in PIC formation and start site selection, and it binds directly to promoter regions [5]. TFIIB is structured in several functional domains, which are named Zn-ribbon, B-reader, B-linker and cyclin folds [5]. The carboxy-terminal domain consists of two cyclin folds, and it is involved in interactions with the promoter and also with TBP, TFIIF and RNAPII, while the amino-terminal zinc ribbon domain interacts with the catalytic center of RNAPII [5]. Furthermore, the amino and carboxy-terminal domains can interact with each other, forming an open or closed conformation. The open conformation is the active form, whereas the closed one is inactive [5]. Thus, transcriptional activators can stabilize the open conformation and therefore can stimulate transcription initiation by allowing PIC formation [4]. Besides the roles in transcription initiation, it is thought that TFIIB also participates in transcription termination [6]. It has been demonstrated that TFIIB is found at the 3′ end of genes, and this is most likely due to the interaction of this factor with termination factors located there [6].

*Pneumocystis* is a group of yeast-like fungus, which can cause pneumonia in AIDS patients and immunocompromised individuals [7]. This fungus is a species-specific pathogen and infects only mammals. Those pathogens seem to be an atypical fungus, since it lacks ergosterol and many biosynthetic pathways are lacking, according to genomic sequencing of the three most studied *Pneumocystis* species: *P. jirovecii* (human), *P. murina* (mice) and *P. carinii* (rat) [8,9,10]. In addition, these species have very few tRNA and rRNA genes and a lack of several metabolic pathways, indicating their high mammal host dependence [11]. The *Pneumocystis* genus differs in the composition of a large gene family encoding the major surface glycoprotein (MSG) or glycoprotein A (gpA) surface antigens [11]. The lack of strains that could be in vitro cultured in large amounts [12] has hampered more detailed molecular and biochemical studies; however, transcriptomics and genomic studies can help to understand the molecular and biochemical processes of this pathogenic fungus [13,14,15]. It is assumed that these organisms possess a conventional transcription apparatus, since from database mining it is possible to find homologous genes encoding RNAPII subunits and GTFs. Furthermore, several transcriptional activators have been described in these organisms [16,17]. Stringer and colleagues have described two TBP genes from rat *P. carinii* and those gene products are highly identical in the conserved core with TBP polypeptides from other eukaryotic species [18]. However, those TBPs have not been characterized further in biochemical or complementation assays. The lack of biochemical and a genetic tractable system has hampered the characterization of genes and gene products from the *Pneumocystis* genus. A biological model organism, relatively genetically close, is the fission yeast *Schizosaccharomyces pombe*, which has a well-studied biochemical and a genetic tractable system [19]. Thus, we think that fission yeast could help to characterize at the genetic, biochemical, and molecular level the genes and gene products from *Pneumocystis* genus. 

Following the above rationale, in this work we have replaced the endogenous fission yeast TFIIB gene by the *Pneumocystis carinii* TFIIB gene in order to gain insights into the function of *Pneumocystis* transcription factors. Our results show that *P. carinii* TFIIB can replace the function of fission yeast TFIIB, although the resulting strain has a slow growth phenotype, and the cell shape is altered. At the molecular level, *Pneumocystis* TFIIB can complement transcription in a TFIIB-depleted fission yeast cell extract, although transcription levels are not as high as those obtained with fission yeast TFIIB. Remarkably, *Pneumocystis* TFIIB cannot support the formation of a promoter–TBP–TFIIB complex but is able to form a promoter–TBP–TFIIB-RNAPII/TFIIF complex. Moreover, primer extension experiments indicate that *Pneumocystis* TFIIB does not change the transcription start site of the *adh1* and *nmt1* fission yeast genes.

## 2. Results

### 2.1. Protein Structures of P. carinii and S. pombe TFIIB Have Conserved Regions

Alignments of the TFIIB amino acid sequences from *S. pombe* and *P. carinii* show a high identity in the described protein domains of TFIIB (Figure 1A). This result suggests high tridimensional (3D) protein structure conservation. To this end, protein modeling was performed using the information of the yeast S. cerevisiae TFIIB 3D structure. The protein 3D structures of TFIIB from *S. pombe* and *P. carinii* showed high homology; however, protein segments at the N-terminal domain and at the beginning of the Cyclin domain 2 showed low identity and some structural differences (Figure 1B,C). The Zn-ribbon domain at the N-terminus is highly conserved and it is able to coordinate a Zn-ion between cysteine residues (Figure 1D).

### 2.2. Characterization of a S. pombe Strain Expressing P. carinii TFIIB

A Clustal alignment of TFIIB polypeptides from budding yeast, human, fission yeast and *Pneumocystis* TFIIB from *P. carinii* shows that *Pneumocystis* TFIIB is genetically closer to fission yeast TFIIB. These results are according to the observations of the protein structures described previously. These observations suggest that *Pneumocystis* TFIIB could replace the essential function of fission yeast TFIIB, since the gene encoding this polypeptide is essential for cell viability. Budding yeast and human TFIIB are distantly related to *Pneumocystis* TFIIB and most likely cannot replace the function of this gene.

Moreover, in budding yeast the TATA boxes are located −120 to −60 nucleotides upstream from the TSS and TFIIB is one of the factors which selects the TSS [20], therefore *Pneumocystis* TFIIB could not replace the function of budding yeast TFIIB. On the other hand, human TFIIB is unable to replace the function of fission yeast to support cell viability, although it can be incorporated into the fission yeast transcription machinery, since when human TFIIB is expressed in fission yeast cells a new TSS appears in the *adh1* gene [20]. Furthermore, human TFIIB cannot support transcription in a reconstituted transcription assay or in TFIIB-depleted cell extracts from fission yeast (see below). 

Next, we sought to replace the chromosomal copy of fission yeast TFIIB by a cDNA encoding *P. carinii* His-tagged TFIIB by gene deletion and replacement. The construct contained the KanMX6 cassette, which confers resistance to the G418 antibiotic (Figure 2A). The transformants were plated on YPD-G418 plates and several transformants were obtained. The deletion of the wild-type copy of TFIIB and its replacement by *Pneumocystis* TFIIB was confirmed by PCR, using genomic DNA from TFIIB-replaced strains (Figure 2B). Protein extracts from the transformants probed with anti-His Tag antibodies show that they contained the His-tagged version of *P. carinii* TFIIB (Figure 2C). The parental control strain does not show any reactivity to the anti-His antibody (Figure 2C). We chose one of these clones for further analysis (clone 3). The resulting strain was named sua7::Pcsua7-kanMX6.

The *P. carinii* TFIIB modified strain grows slower than the parental wild-type strain in YPD and minimal liquid media (Figure 2D) and in YPD-agar plates (Figure 2E). The cell morphology of this strain is altered, since the cells from the *P. carinii* TFIIB-modified strain are apparently longer than wild-type cells (Figure 2E,F). Mutant cells showed a significant increase in the length compared with wild-type cells (Figure 2G).

These observations, and the fact that TFIIB was exchanged, might indicate that there is a defect in transcription of several endogenous genes which are necessary for normal cell growth.

### 2.3. Pneumocystis TFIIB Cannot Support the Formation of a Promoter–TBP–TFIIB Complex but Is Able to Form a Promoter–TBP–TFIIB-RNAPII/TFIIF Complex

To study the molecular functions of *P. carinii* TFIIB, the polypeptide was expressed in *E. coli* and purified by Ni-NTA-agarose affinity chromatography (Figure 3A). The polypeptide has an apparent molecular weight of 32 KDa and migrates closer to fission yeast TFIIB (Figure 3B). Analysis by SDS-PAGE of the recombinant TFIIB polypeptides indicates that both *S. pombe* and *P. carnii* proteins are more than 90% pure as judged by Coomassie Blue R-250 staining (Figure 3A,B). These results are in agreement with the expected size of *P. carinii* TFIIB (337 aa) and fission yeast TFIIB (340 aa). Both polypeptides react with anti-His tag antibodies, confirming the identity of those polypeptides (Figure 3C).

To further study the assembly of a PIC on a TATA box promoter using *P. carinii* TFIIB and recombinant fission yeast transcription factors together with RNAPII, we used electrophoretic mobility shift assays (EMSA). It is well established that the binding of TBP to the TATA box is stabilized by TFIIB. The results shown in Figure 3D indicate that *P. carinii* TFIIB is unable to interact with fission yeast TBP to form a strong promoter–TBP–TFIIB complex (Figure 3D, lane 2). This result is surprising since fission yeast TFIIB can easily form that complex (Figure 3D, lane 2). This indicates that the *P. carinii* TFIIB interaction with fission yeast TBP is rather weak. Remarkably, the addition of TFIIF/RNAPII to the promoter–TBP mixture together with *P. carinii* TFIIB can form a new complex on the promoter DNA, which is similar to that formed with fission yeast TFIIB (Figure 3D, lanes 4, 8 and 9). The presence of RNAPII in the complex was determined using the antibody SWG16 (anti-CTD) which was added to the binding mix and the complex was dissociated (Figure 3D, lane 5). In addition, TBP-RNAPII/TFIIF are unable to form a complex on the promoter (Figure 3D, lane 6). These results suggest that TFIIF/RNAPII can promote the formation of the promoter–TBP–TFIIB complex in a cooperative fashion. Afterwards, TFIIE can bind to the promoter–TBP–TFIIB–TFIIF/RNAPII complex as it is seen in the PIC formation on a TATA box promoter using human RNAPII and recombinant transcription factors (Figure 3D, lane 10).

### 2.4. The Expression Levels of TBP, TFIIF and RNAPII Do Not Change by the Replacement of Fission Yeast TFIIB by P. carinii TFIIB

We studied whether the expression levels of GTFs and RNAPII can be altered by gene replacement. Western blot analysis of cell extracts from wild-type (wt) and TFIIB-replaced (mut) strains indicate that the levels in both strains are the same (Figure 4). TBP levels remain the same in both strains (Figure 4A,D) as well as the levels of the largest subunit of RNAPII (Rpb1, Figure 4B,E). Furthermore, the level of TFIIF (TFIIFβ) was not altered by the TFIIB replacement (Figure 4C,F). These results indicate that the GTFs and RNAPII are expressed normally in the TFIIB-replaced strain.

### 2.5. Pneumocystis TFIIB Can Complement Transcription in a TFIIB-Depleted Fission Yeast Cell Extract

Next, we studied the transcriptional activity of *Pneumocystis* TFIIB in a fission yeast TFIIB-depleted cell extract. Fission yeast TFIIB was depleted from the cell extract using antibodies anti-TFIIB crosslinked to Sepharose beads. The antibodies were able to deplete more than 90% of the fission yeast TFIIB from the cell extract as detected by Western blot anti-TFIIB (Figure 5A,B). However, there was no noticeable depletion of two factors, namely TBP and TFIIFβ (Figure 5A,B). Furthermore, RNAPII was not depleted either by the anti-TFIIB antibodies (Figure 5A,B). As expected, the TFIIB-depleted extract was inactive in in vitro transcription, but the transcription was rescued by adding back fission yeast TFIIB (Figure 5C,D). When increasing amounts of *Pneumocystis* TFIIB were added to the TFIIB-depleted cell extract, the transcription can be recovered, although at lower levels than with fission yeast TFIIB (Figure 5C,D). These results indicate that *Pneumocystis* TFIIB can replace fission yeast TFIIB in in vitro transcription, although lower levels are obtained compared with fission yeast TFIIB. The results shown indicate that 50% of the transcriptional levels are recovered with 20 ng of *P. carinii* TFIIB versus 80% with the same amount of fission yeast TFIIB, compared with WCE (100% transcriptional level) as control (Figure 5D).

### 2.6. Pneumocystis TFIIB Does Not Change the Transcription Start Site of the nmt1 and adh1 Fission Yeast Genes within a Narrow Window

The TSSs of the *nmt1* and *adh1* have been reported in vivo as well as in vitro [20] and they are summarized in Figure 6A. To determine whether the TSS of the endogenous genes could be changed in vivo by the replacement of fission yeast TFIIB by *Pneumocystis* TFIIB, we isolated RNA from the wild-type and the replaced strain and the TSSs of the *nmt1* and *adh1* genes were determined by primer extension. Two TSS have been reported in the *nmt1* gene (see Figure 6A) and the results showed that the TSSs of the *nmt1* gene do not change in the replaced strain compared to the wild-type strain, although the expression levels are lower in the replaced strain (Figure 6B,C). This finding would suggest that some genes are poorly expressed in the TFIIB-replaced strain. In the TFIIB-replaced strain, the major transcription start site of the *adh1* gene is slightly offset from the major start site of the wild-type strain, suggesting that in the TFIIB-replaced strain the transcription starts 2–3 nucleotides further downstream from +1 of the wild-type strain, but always in purines (A or G, see Figure 6A–C). Furthermore, in the TFIIB-replaced strain a TSS downstream of the major TSS is used; however, this TSS is poorly used in the wild-type strain (Figure 6B,C). These results demonstrate that the TSS of the *adh1* gene promoter does not change within a narrow window (20–40 bp) from the edge of the TATA element as has been reported earlier.

## 3. Discussion

*Pneumocystis* genus is an opportunistic fungal pathogen prevalent in infections in patients with AIDS and in immunocompromised individuals. The analysis of this pathogen has been hampered by the inability to isolate it in a pure culture; however, the use of modern molecular techniques and genomic analysis has shed light on its complex cell biology [7,21]. Key proteins involved in the mitotic cell cycle, signal transduction pathways, cell wall assembly and metabolic pathways have been identified and functional analysis of many genes encoding proteins has been done using *S. cerevisiae* and *S. pombe*, which are phylogenetically related to *Pneumocystis* [14]. Most of the expressed sequence tags (ESTs) identified in the genome sequencing project of *P. carinii* are homologous to *S. cerevisiae* and *S. pombe* ESTs [14]. Even so, the function of *Pneumocystis carinii* genes cannot be assigned based on homology alone. 

The transcriptional machinery of *Pneumocystis* has not been studied yet, although it is assumed that *P. carinii* has its own conventional transcriptional machinery. BLASTP searches in the *Pneumocystis* database at the Genbank at the NCBI indicate that there exist genes encoding polypeptides for RNAPI, RNAPII and RNAPIII. Furthermore, it is possible to find genes encoding all the GTFs for RNAPII transcription. Moreover, two genes encoding TBP have been cloned from *P. carinii* [18]. However, those genes have not been functionally characterized. Thus, a biological system is needed to characterize the genes involved in messenger RNA transcription.

Since there is not a *Pneumocystis* promoter database, very little is known about gene expression and the core promoter elements that regulate transcription initiation in those pathogens. A few genes and their promoter sequences have been described in *Pneumocystis*. For example, the actin-encoding gene from *P. carinii* has been described and at its TSS has been determined [22,23]. A putative TATA box was found at −39 relative to the major TSS. The *arom* gene from *P. carinii* has also been isolated and the 5′ untranslated sequence contains a TATA box at −86 to −82 relative to the 5′ methionine initiation codon [24,25]. MSG is the most abundant surface protein in *Pneumocystis* and plays an important role in the interaction of this pathogen with the host cells since it potentially facilitates evasion of the host immune response by antigenic variation [26]. MSG is encoded by a multicopy family of related genes, which is estimated in 25–80 copies by genome, and clustered in tandem arrays near the telomere of most of the chromosomes [27]. However, available evidence suggests that only one MSG gene is expressed in an individual cell and the expressed MSG gene is located downstream of a region, which is named the upstream conserved sequence (UCS) [28,29]. In a study aimed to identify promoter regions important for MSG expression, a heterologous system (*S. cerevisiae*) was used [30]. The results indicate that a 1.4 kb sequence upstream of the ATG start codon of *P. carinii* MSG gene had promoter activity in *S. cerevisiae* cells and a 13 bp sequence from −116 to −104, relative to the ATG start codon was important for MSG promoter activity [30]. This sequence CATATTAAGGGGA is important for promoter activity and also putative TATA boxes were identified in *Pneumocystis* MSG genes, which are located at −30 from the TSS. We believe that most likely the TATA boxes in *Pneumocystis* are located around −30 relative to the TSS as they are located in higher eukaryotes and *S. pombe* promoters.

It is known that TFIIB and RNAPII can participate in transcription start site selection and TFIIB plays a pivotal role in PIC formation, since it bridges promoter-bound TBP and TFIIF/RNAPII [5,31,32,33,34,35]. In this regard, the TSS of the endogenous *S. pombe adh1* gene does not change within a narrow window of approximately 20–40 bp from the edge of the TATA element, when *P. carinii* TFIIB replaces the *S. pombe* TFIIB. This indicates that *P. carinii* TFIIB can work correctly in start site selection together with *S. pombe* RNAPII. Moreover, the TSS of the *S. pombe nmt1* gene does not change either, although it seems to be that it is poorly expressed when *P. carinii* TFIIB replaces the wild-type TFIIB. In an in vitro transcription assay, the *S. pombe adh1* gene initiates at three sites, similar to what is seen in vivo [20]. However, when human TFIIB protein is overexpressed in *S. pombe* cells, the main TSS of the *adh1* gene is not altered; however, a new transcription site appears which is closer to the TATA box [20]. This indicates that in vivo and at the *adh1* promoter, human TFIIB can form a PIC and redirect *S. pombe* RNAPII at a different TSS. However, whether or not human TFIIB can replace *S. pombe* TFIIB function has not been formally tested yet. In our laboratory, the attempts to replace in vitro *S. pombe* TFIIB by its human counterpart have failed. This evidence indicates that perhaps human TFIIB only works at a small set of promoters in vivo.

Taken altogether, our results indicate that *P. carinii* TFIIB can replace in vivo and in vitro the function of *S. pombe* TFIIB and this model can be useful to characterize the function of the RNAPII transcription machinery from *Pneumocystis* to deeply understand the transcription process in this opportunistic pathogen. 

## 4. Materials and Methods

### 4.1. Sequences and Protein Modeling

Fission yeast *S. pombe* TFIIB nucleotide (NM_001019652.2) and protein (NP_594229.1) sequences were used as queries to search for *P. carinii* TFIIB sequences using the online BLAST tool at the Genbank in the NCBI. In this search we found sequences with high identity (over 60% identity), namely XM_018369306.1 (nucleotide) and XP_018227014.1 (protein). The building of the protein structure models was performed using the online Structure Homology-modeling tool (available at https//:swissmodel.expasy.org, accessed on 30 May 2022). The amino acid sequences of *P. carinii* and fission yeast TFIIB were used as input to build the models using the budding yeast TFIIB template 7o4i.1.M. Models were generated with a sequence identity near to 40%.

### 4.2. Yeast Strains and Media

The fission yeast *S. pombe* wild-type strain LP32 (h^−^ Sleu1-32) was used to perform the TFIIB gene replacement. *S. pombe* cells were grown in YPD medium or minimal medium (EMM, US Biological, Salem, MA, USA). For growth analysis, cells were grown for 16 h at 30 °C in a shaker until OD_600_ = 1.8. Afterwards, 20 μL of each culture was mixed with medium YPD or EMM to give a final volume of 200 μL and then put in a 96-well plate and incubated for 24 h at 30 °C in an Infinite 200 PRO equipment (Tecan, Männedorf, Switzerland). After the culture reached stationary phase, an aliquot of each culture was analyzed by microscopy and the length of 25 cells per field were evaluated using Image J software (NIH, Bethesda, MD, USA). Microdilution assays were performed from saturated yeast cultures grown overnight at 30 °C. Cultures were adjusted to OD_600_ = 0.1 and dilutions were performed at 1/10 and 1/100. A total of 2 μL of each dilution was plated in YPD-agar and grown for 48 h at 30 °C.

### 4.3. Gene Replacement

To replace the sequence of fission yeast TFIIB gene (*sua7*) by the *P. carinii* homologue sequence, a deep search at the Genbank was performed. By searching in the *P. carinii* genome, the TFIIB ORF and the flanking regions were extracted. Using an in silico approach to the ORF, a His-tag and stop codon were added, followed by a KanMX6 cassette as a selectable marker. Contiguous to the KanMX6 cassette, the 3′ flanking region of the fission yeast TFIIB gene was incorporated (257 bp length). At the 5′ of the ORF, a 5′ flanking sequence from the fission yeast TFIIB gene was added (214 bp length). The entire nucleotide sequence was synthesized at GenScript (Piscataway, NJ, USA) and inserted into the pUC57 vector in the EcoRV restriction site. Using standard M13 (forward and reverse) primers, the complete sequence was amplified by PCR generating a 3000 bp DNA product. The amplified DNA fragment was purified and resuspended in nuclease-free water at 50 ng/μL. Afterwards, 50 ng of the PCR product was used to transform lithium acetate competent *S. pombe* cells (LP36 strain; h^−^ Sleu1-32). Transformed cells were recovered at 4 h at 30 °C in recovery media and then were plated on YPD-agar supplemented with 200 μg/mL G418 (Sigma-Aldrich, St. Louis, MO, USA) to select the transformants. Resistant colonies appeared after 3–4 days and those were grown in YPD liquid media supplemented with G418 and analyzed for the insertion of the His-tagged *P. carinii* TFIIB in cell extracts. Glycerol stocks of those strains were stored at −80 °C until use. The TFIIB-replaced strain generated was named sua7::sua7Pc-KanMX6, usually called mutant strain (mut).

### 4.4. Pneumocystis TFIIB Expression and Purification in E. coli

A cDNA encoding the *P. carinii* TFIIB ORF was synthesized at Genscript (Piscataway, NJ, USA) and inserted into the NdeI and BamHI restriction sites of pET-15b (Novagen, Madison, WI, USA). This expression vector was transformed in *E. coli* BL21(DE3) and plated on LB-agar supplemented with ampicillin. The resulting colonies were grown in LB liquid media supplemented with ampicillin until O.D._600_ 0.8. Then, the recombinant protein production in the culture was induced by addition of 0.5 mM isopropyl β-D-1-thiogalactopyranoside (IPTG, Sigma-Aldrich, St. Louis, MO, USA) for 3 h. Cells were harvested and processed as described elsewhere [36]. TFIIB was purified from the inclusion bodies as described for fission yeast TFIIB [37]. The last step of purification of *P. carinii* TFIIB was done by using a Ni-NTA-Agarose column. Proteins were eluted from the column by an Imidazole concentration gradient. Fractions with high amounts of TFIIB protein were analyzed and detected by SDS-PAGE followed by Coomassie blue R-250 staining and Western blot analysis using anti-His tag monoclonal antibodies. Fractions containing TFIIB were pooled, quantified and stored at −80 °C until use.

### 4.5. S. pombe RNAPII and GTFs Purification

Recombinant *S. pombe* GTFs (TFIIA, TBP, TFIIB, TFIIE and TFIIF) were expressed and purified as described previously [36]. RNAPII core enzyme was purified from a wild-type *S. pombe* strain using conventional chromatography as described previously [36].

### 4.6. Electrophoretic Mobility Shift Assay (EMSA)

EMSA assays were performed essentially as described previously [37,38]. Binding reactions contained binding mix: 20 mM HEPES (pH 7.9), 50 mM KCl, 5 mM MgCl_2_, 0.1 mM EDTA, 5% glycerol, 0.5% PEG 8000 (Sigma-Aldrich, St Louis, MO, USA), 2 mM DTT, 0.1 mM PMSF, 50 ng acetylated BSA and 50 ng poly(dG-dC). Purified recombinant *S. pombe* GTFs, *P. carinii* TFIIB and *S. pombe* RNAPII were used in the assays. Proteins were incubated with a binding mix for 5 min at 25 °C. Then, 5–10 ng of double stranded Ad-MLP TATA box 5′ end ^32^P-labeled probe (5′ GGCTATAAAAGGGGGTGGGGAGGACTTCCCCCCGATATTTTCCCCCACCCC 3′) was added to the assays. Reactions were incubated for 15 min at 30 °C. DNA–protein complexes were analyzed in 5% polyacrylamide gels containing 10% glycerol, 0.5X TBE and run at 100 V at 4 °C for 2 h. Detection of the complexes on the gels was performed by autoradiography analysis.

### 4.7. TFIIB Depletion

TFIIB was depleted from a *S. pombe* wild-type WCE by adjusting the WCE at 0.5 M potassium acetate and then incubating with a column that contained 1 mg/mL of rabbit anti-TFIIB antibodies crosslinked to CNBr-activated Sepharose. A total of 1 mL of WCE was incubated with 1 mL of settled resine. The mix was incubated by rocking for 30 min at room temperature and then for two hours at 4 °C. The mix was centrifuged and the supernatant used as a TFIIB-depleted WCE (DWCE).

### 4.8. In Vitro Transcription Assay

In vitro transcription was performed as described previously [36,38] using 100 ng of plasmid DNA containing the Ad-MLP promoter fused to the G-less cassette. The plasmid DNA used for transcription reactions was purified by the E.Z.N.A. Plasmid Midi kit according to manufacturer’s indications (Omega Bio-tek, Norcross, GA, USA). The promoter was fused to the G-less cassette and upon digestion of the transcripts by RNAase T1 produced a transcript of approximately 370 nucleotides. Reactions were performed with 5 μL of *S. pombe* WCE (10 mg/mL) or DWCE (TFIIB-depleted extract at 10 mg/mL). Detection of transcripts on the gels was performed by autoradiography analysis. 

### 4.9. Primer Extension

Wild-type *S. pombe* and TFIIB-replaced strains were grown in liquid minimal media, without thiamine. Cells were collected, washed twice with nuclease free water and then resuspended in 10 mM Tris pH 7.8 and 1 mM EDTA (TE buffer), and then one volume of acid-washed glass beads was added. The mix was vortexed for minutes and after centrifuged at 12,000 rpm at 4 °C for ten minutes. Total RNA from the supernatant was purified by using the RNeasy mini kit (Qiagen, Hilden, Germany) and 1 μg of total RNA was used for primer extension reactions. The primer extension reactions were performed using M-MLV reverse transcriptase (Promega, Wisconsin, WI, USA) according to the manufacturer instructions and the primer extended products were separated in a 10% polyacrylamide-urea gel. To determinate the 5′ ends and the transcripts of the *adh1* gene, a 5′ end ^32^P-labeled specific primer (5′ AGGTACTGCGGAATTCGGTTG 3′) was used, which is complementary to bases +181 to +200, where +1 is the first nucleotide of the mRNA of locus NM_001023338 (Genbank Accession Number). To determine the 5′ ends and transcripts of the nmt1 gene, a 5′ end ^32^P-labeled specific primer (5′ TCACTTTCCTCACAAACTGG 3′) was used, complementary to bases + 17 to +36, where +1 is the ATG translation initiation codon. 

### 4.10. Statistics

GraphPad Prism 9 (GraphPad Software Inc., San Diego, CA, USA) was used for all the analyses. Results were expressed as the mean ± standard deviation (SD). All experiments were repeated three times. Analysis of autoradiography films was performed using the Image J software (NIH, Bethesda, MD, USA) Distribution of data was evaluated by Shapiro–Wilk test and differences between experimental groups were evaluated using Student’s *t*-test. *p* < 0.05 was considered as statistically significant.

## Figures and Tables

**Figure 1 ijms-23-06865-f001:**
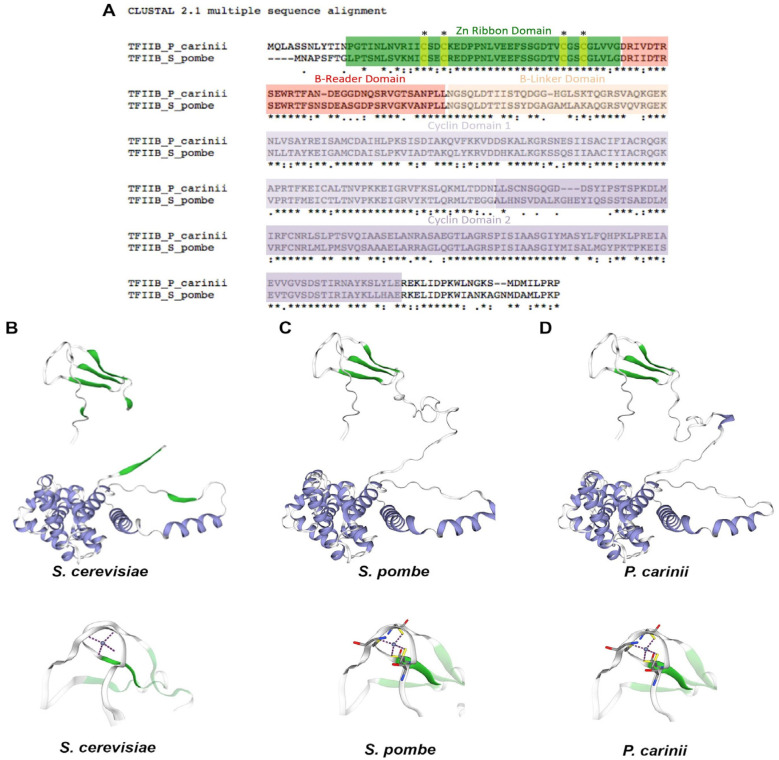
***P. carinii* TFIIB protein has highly conserved regions.** (**A**) Comparison of TFIIB amino acid sequences from *P. carinii* and *S. pombe* (fission yeast) TFIIB. In colors are highlighted the known TFIIB domains and names are given over each domain. The alignment was performed with the ClustalW program (https://www.genome.jp/tools-bin/clustalw, accessed on 21 March 2022). (**B**) 3D structure of TFIIB from *S. cerevisiae*, which was the template to generate structures of panels C and D. (**C**,**D**) Predicted 3D structures of TFIIB from *S. pombe* and *P. carinii* were determined using the online tool Structure homology-modeling (https://swissmodel.expasy.org, accessed on 30 May 2022) using as a template the crystal structure information of budding yeast TFIIB. In the 3D structures, the Zn-ribbon domain is highlighted in green and cyclin domains are highlighted in purple. Bottom panels show a modeling of the Zn-ribbon domain of *S. cerevisiae*, fission yeast and *P. carinii* TFIIB that shows the Zn ion (blue) coordinated by cysteine residues (yellow).

**Figure 2 ijms-23-06865-f002:**
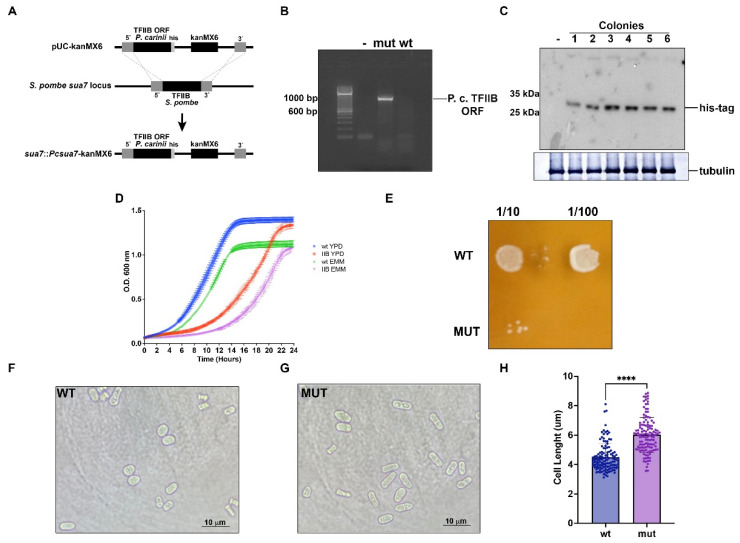
**Characterization of a sua7::Pcsua7-kanMX6 strain.** (**A**) Schematic representation of the construction used for *S. pombe* TFIIB (SpIIB) deletion and replacement by *P. carinii* TFIIB (PcIIB). The construct was made as described in Materials and Methods and amplified by PCR. The PCR products were used to transform *S. pombe* cells and simultaneously delete and replace the endogenous SpIIB (sua7) gene by a cDNA encoding the PcIIB. Resulting strain was named sua7::Pcsua7-kanMX6. (**B**) PCR analysis of genomic DNA from mutant and wild-type cells, indicating the presence of the *P. carinii* TFIIB ORF. (**C**) Resulting colonies were resistant to G148 antibiotic, grew in liquid cultures in YPD media and analyzed further by the presence of the His-tagged PcIIB by Western blot. All analyzed colonies from 1–6 possessed the His-tagged PcTFIIB and colony 3 was chosen for further studies—indicates wild-type cell proteins. Bottom panel indicates the presence of tubulin as internal control. (**D**) Wild-type cells (WT) cells and sua7::Pcsua7-kanMX6 (IIB) cells containing PcIIB were grown during 24 h in EMM and YPD liquid media and the growth data were recorded. (**E**) Microdilution analysis of colonies from wild-type and mutant strains; 1/10 and 1/100 dilution from saturated cultures growth in YPD medium were plated in YPD agar. Colonies were grown at 30 °C and were evaluated after 48 h. Wild-type (**F**, WT) and mutant (**G**, MUT) cells grown in YPD medium were observed by microscopy using a magnification of 1000X to evaluate their shape. (**H**) Quantification of cell length. Analyses were performed measuring the length of 25 cells per field using 1000× of magnification and 5 fields. **** indicates *p* < 0.0001.

**Figure 3 ijms-23-06865-f003:**
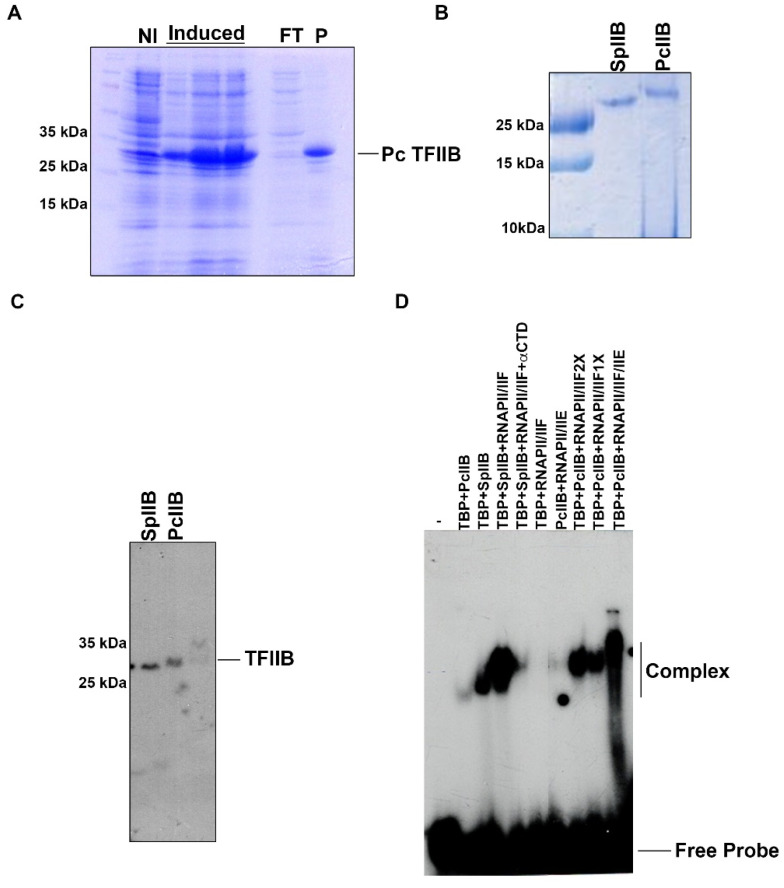
**Purification of recombinant *P. carinii* TFIIB and complex formation on a TATA box promoter.** (**A**) Induction and purification of PcIIB. A pET-15b construct carrying on the PcIIB cDNA was transformed in *E. coli* BL21(DE3) and induced with IPTG to produce the recombinant PcIIB. The transformed *E. coli* cells were grown at OD_600_ of 0.6 and induced with IPTG for 1, 2 and 4 h. The non-induced (NI) and induced cells were broken by SDS sample buffer and analyzed for the induced polypeptide in a 10% SDS-PAGE gel followed by Coomassie Blue R-250 staining. Note that the NI culture produces low amounts of the *P. carinii* TFIIB in the absence of IPTG. This is due to traces of inducers in the LB media. The recombinant protein was insoluble, and it was purified from the pellets as described in Materials and Methods. Lane FT is the proteins that do not bind to the NTA-Ni-Agarose column and P represents the bound proteins to the column. (**B**) A 10% SDS-PAGE gel to compare and analyze *S. pombe* TFIIB (SpIIB) and *P. carinii* TFIIB (PcIIB). (**C**) Western blot to analyze recombinant TFIIB. The samples were separated on a 10% SDS-PAGE and transferred to PDVF membranes and probed with anti-His tag monoclonal antibodies. (**D**) Complex formation of PcIIB and fission yeast TBP, TFIIF, RNAPII and TFIIE on the Ad-MLP TATA box. An EMSA was performed using different combinations of GTFs and RNAPII as indicated at the top of the figure. The amounts of PcIIB, TBP, SpIIB and TFIIE were 20 ng of each factor in the reaction. TFIIF1X represents 20 ng and TFIIF2X represents 40 ng of the added factor. The used amount of RNAPII was 100 ng.

**Figure 4 ijms-23-06865-f004:**
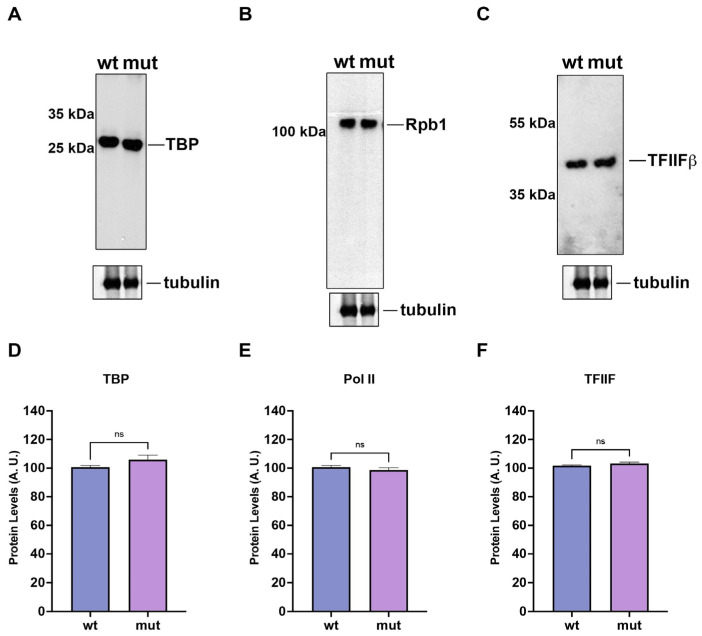
**Transcription factors TBP, TFIIF and RNAPII levels do not vary in the fission yeast sua7::Pcsua7-kanMX6 strain.** Fission yeast whole cell extracts were prepared from wild-type (wt) and TFIIB-replaced (mut) strains and the levels of the transcription factors and RNAPII were determined by Western blots using 2 μg of whole cell extract and anti-TBP (**A**), anti-CTD (**B**) and anti-TFIIFβ (**C**) antibodies. Bottom panels indicate the presence of tubulin as internal control. Quantification of Western blots of TBP (**D**), RNAPII (**E**) and TFIIF (**F**) are indicated. ns indicates non-significant differences after Student’s *t*-test.

**Figure 5 ijms-23-06865-f005:**
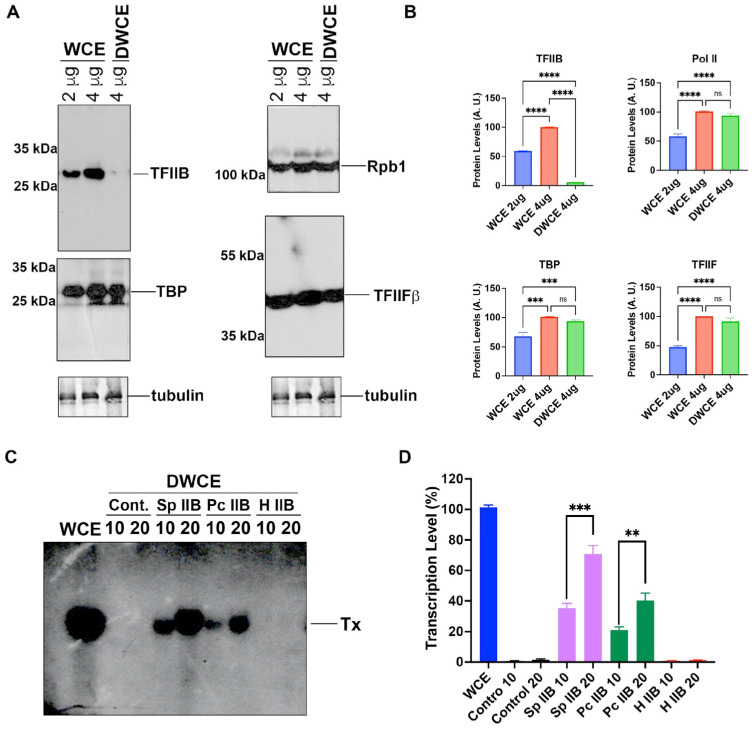
***P. carinii* TFIIB can replace in vitro the transcriptional functions of *S. pombe* TFIIB.** (**A**) A whole cell extract was prepared from fission yeast and depleted of TFIIB by using antibody-affinity chromatography as described in Materials and Methods. A mock depleted whole cell extract (WCE) and a TFIIB-depleted whole cell extract (DWCE) were analyzed by Western blot using anti-*S. pombe* TFIIB antibodies cross linked to Sepharose beads. We analyzed 2 and 4 μg of WCE and 4 μg of DWCE. The same amounts of protein were analyzed for TBP, TFIIFβ and RNAPII in separate Western blots. Tubulin analyses are indicated as internal control. (**B**) Quantification of Western blot. ns indicates non significative differences; *** indicates *p* < 0.001; **** indicates *p* < 0.0001 after Student’s *t*-test. (**C**) Transcription assay and recovery of the transcriptional activity by different TFIIBs. The DWCE was supplemented with 10, 20 ng of heat-inactivated *S. pombe* TFIIB (control) and with 10, 20 ng of active *S. pombe* TFIIB (Sp IIB). The DWCE was supplemented with 10, 20 ng of active *P. carinii* TFIIB (Pc IIB). Human TFIIB (H IIB) was also used to supplement the DWCE (10 and 20 ng). (**D**) Quantification of Figure 5C. ** indicates *p* < 0.005 and *** indicates *p* < 0.001 after Student’s *t*-test.

**Figure 6 ijms-23-06865-f006:**
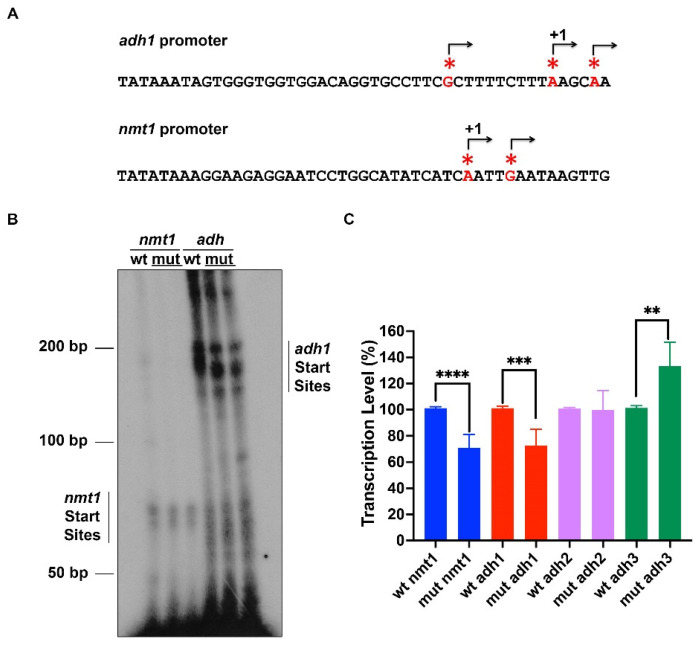
**Replacement of SpIIB by PcIIB does not change the in vivo TSS in the *nmt1* and *adh1* gene promoters.** (**A**) In vitro and in vivo TSS from the *adh1* and *nmt1* genes. Red asterixes and arrows indicate the TSS of each gene. +1 indicates the major TSS of *adh1* and *nmt1* gene promoters. (**B**) Total RNA was isolated from wild-type and PcIIB-replaced (mut) fission yeast cells, which were grown in minimal medium. The RNA (1 μg) was analyzed by primer extension using a radiolabeled oligonucleotide and the products were analyzed using a 10% polyacrylamide-urea gel, followed by autoradiography. The products indicated by a bar are correctly initiated transcripts from the *nmt1* and *adh1* gene promoters. The wild-type strain and two different clones of PcIIB-replaced strain were analyzed (mut). (**C**) Quantification of the start sites is indicated. Three adh start sites were analyzed from the top (adh1) to the bottom (adh3). Comparison of transcription levels was performed using a Student’s *t*-test. ** indicates *p* < 0.005; *** indicates *p* < 0.001; **** indicates *p* < 0.0001.

## Data Availability

Not applicable.

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
