# Peer review of "RNA Polymerase II Transcription in Pneumocystis: TFIIB from Pneumocystis carinii Can Replace the Transcriptional Functions of Fission Yeast Schizosaccharomyces pombe TFIIB In Vivo and In Vitro"

_ijms, 2022, doi:10.3390/ijms23126865_

Round 1

Reviewer 1 Report

Dear authors, 

As you also pointed out, the unfeasibility to grow this fungus in vitro has hampered the studies so far possible, thus most of the knowledge achieved in other pathogenic fungal species is not yet available in Pneumocystis. Even though genomics led to major advances in comprehend its biology and physiology or rather to speculate about them, classical molecular techniques ensure further steps in this direction. Complementation studies of homologues genes such as yours here presented are an example. The transcriptional machinery is one of the unexplored topic and the present work is therefore novel. The experimental design and methods used are appropriate. The significant results are presented in a very clear manner to the reader and they are scientifically solid and support the conclusion that were reached in the discussion. According to my revision, the results can be published in the current form and will be of great interest for the Pneumocystis scientific community.

Best Regards. 

Author Response

Reply to the reviewers:

First, we would like to thank both reviewers for the careful and critical revision of our manuscript. Those comments will certainly improve the contents of the submitted manuscript.  Below you can find the answers to each comment, and we hope that now the manuscript would be ready for publication.

Reviewer 1:

  • Thanks to reviewer 1 for the optimal scientific evaluation of our manuscript. We believe that this work will be a contribution to the Pneumocystis

Reviewer 2 Report

Summary:

The Pneumocystis genus is an opportunistic fungal pathogen that infects patients with AIDS and immunocompromised individuals. This pathogen is difficult to study because it cannot be grown in culture. Due to this issue, transcription has not been studied in the Pneumocystis genus. The goal of this work was to characterize the function of the RNAPII general transcription factor TFIIB from Pneumocystis carinii using the model organism S. pombe. The results from this body of work demonstrate that PcTFIIB can replace the essential function of SpTFIIB both in in vivo and in vitro assays. While PcTFIIB promotes transcription in S. pombe, there are growth and genetic defects that demonstrate that there are likely significant differences in transcription when PcTFIIB replaces SpTFIIB.

General Comments:

  1. While the authors demonstrate that PcTFIIB can replace SpTFIIB, they also demonstrate that the replacement causes some changes in gene expression and does not fully recapitulate the function of SpTFIIB. This suggests that it would be difficult to directly study the function of the Pneumocystis Pol II system in pombe. Since replacement of SpTFIIB with PcTFIIB does partially rescue Pol II transcription, this system could potentially be used to test whether certain drugs blocked the function of PcTFIIB by using transcription as a read out. Additionally, this system could be used to investigate broad functions of different proteins, but any specific functions would have to be studied in a system that is more related to Pneumocystis.
  2. This manuscript needs to be heavily revised to grammatical errors. There are numerous sentences that are confusing and numerous grammatical mistakes. Additionally, the results section needs to be edited to include italicizing all genus species names.
  3. Since we already know that TFIIB is a required GTF, this work does not significantly add to the field of understanding RNAPII transcription. It does tell us that TFIIB is not exactly species specific and TFIIB from other closely related organisms could probably be used to replace TFIIB in Sp. But one could hypothesize that RNAPII transcription in fungi would be very similar. Additionally, no unique function of TFIIB was described.

Specific Comments:

Introduction: Overall, the introduction highlights the background of the topic and describes the rationale of the work. In paragraph 2 of the introduction, the authors discuss that TFIIB can form two conformations: a closed (inactive) or open (active) conformation. They state that transcriptional activators can stabilize the open conformation, thereby stimulating transcription. Questions I had as I read through this paragraph were: 1) is TFIIB naturally in an inactive conformation or is it constitutively active? Additionally, are the transcriptional activators that stabilize TFIIB in Pc present in Sp or vice versa. If some of this information is known, it would be beneficial to include some of this information in the introduction. Knowing whether PcTFIIB needs to have activators to stabilize the open conformation or not and whether they are present in Sc would help solidify the rationale for using Sp as a model organize to study Pol II transcription in Pc.

Figure 1:

  • The colors used in Panel A should translate to the colors used in Panels B-D. For example, the Zn Ribbon Domain is highlighted in an orange color in Panel A. This domain should be colored orange in Panels Band C.
  • It would be beneficial to show the solved structure of cerevisiae TFIIB since that structure was used as a model for the predicted structures in Panels B-D.

Figure 2:

  • The PCR results that confirm the deletion of SpTFIIB and replacement by PcTFIIB should be shown in this figure, discussed in lines 151-153.
  • On Panel B, there needs to be an internal standard on the Western blot, e. house-keeping protein like actin to show that equal amounts of protein were loaded in each lane.
  • Images of colony growth discussed in lines 156-157 should be shown.
  • The authors claim that there is a morphology change in the mutant yeast where the yeast cells are longer. They only show two representative images to demonstrate this change in morphology. The authors should quantify this change with ImageJ or a similar software and graph their results to see if there is a significant change in cell length.

Figure 3:

  • First, it is not clear on whether the authors purified just the PcTFIIB protein from coli or if they also purified SpTFIIB from yeast.
  • Panel A only shows the results from the PcTFIIB purification.
  • Line 181 refers to the protein reacting with the anti-His antibody for Panel B, but panel B is a Coomassie stained PAGE gel, so it is not clear on what the authors are referring to.
  • Line 181-183 refers to both proteins being more than 90% pure but results for only 1 protein (PcTFIIB) are shown and no analysis was indicated.
  • Figure legend line 211 – change 11% to 10%
  • Panel D:
    • What is the difference between IIF and IIF 1X?
    • The authors did not perform other experiments to demonstrate that the TBP+IIB+RNAPII/IIF complex was forming. RNAPII is a large protein complex. When bound to DNA+TBP+IIB, this should result in a sizeable shift. However, the shift + Pol II is not that big (Comparing lanes 3 and 4 of the EMSA.). The authors need to provide some evidence that Pol II is in the complex.
    • The overall quality of the EMSA should be improved. It is worrisome that so much excess DNA is needed to get a good band shift that demonstrates protein-DNA interaction.
    • The data displayed in Panel D does not fully support the author’s conclusions. Experiments need to be done to demonstrate that RNAPII requires association of TBP+TFIIB to bind to the DNA. Additionally, experiments should be performed that demonstrate TFIIE binds to the complex in lane 6 and that the entire complex is required for TFIIE to bind.
    • A lot of conclusions on binding and complex formation are made from this one EMSA. More experiments need to be performed to solidify the author’s conclusions.
    • This experiment requires additional controls.
  • Panel B: what is the purpose of Pane l B?

Figure 4:

  • Each of these WB’s should have a internal standard for load (actin or tubulin).
  • These WB’s should be quantified.

Figure 5:

  • No internal for the WB’s.
  • No statistics were shown comparing SpTFIIB and PcTFIIB. Is there any statistical difference?

Figure 6:

  • No major issues with Figure 6.

Author Response

First, we would like to thank both reviewers for the careful and critical revision of our manuscript. Those comments will certainly improve the contents of the submitted manuscript.  Below you can find the answers to each comment, and we hope that now the manuscript would be ready for publication.

Reviewer 2

We thanks to reviewer 2 for the raised comments and concerns. Please, you can find below the answers to your comments and concerns.

  • General comment 1. It is true that PcTFIIB does not fully complement the in vivo functions of SpTFIIB, suggesting that the replacement might produce an effect on gene expression. However, we cannot conclude that other components of the RNAPII transcription machinery from Pneumocystis would not fully complement the function of their orthologues in fission yeast. This has not been tested yet, and it is possible that genes encoding subunits of TFIIF, TFIIE and RNAPII from Pneumocystis could fully replace the essential functions of those genes and they can support transcription. Moreover, heterologous in vitro transcription systems could be less restrictive than in vivo assays and they could be used to investigate the functions of transcription factors from Pneumocystis. Besides, as pointed out by the reviewer, this assay can be used to test for drugs that can block the function of PcTFIIB and moreover, to obtain suppressor genes from fission yeast that could restore the normal growth.

  • General comment 2. We have revised and corrected all grammatical mistakes.

  • General comment 3. Yes, we know that TFIIB is a required GTF in all eukaryotic models studied so far, however, we did not know that PcTFIIB had transcriptional functions. TFIIB seems not to be species specific, but it needs to be closely related to replace the essential functions, since neither human nor budding yeast TFIIB can replace SpTFIIB transcriptional functions. It is true that we can hypothesize that RNAPII transcription from all fungi would very similar, however, we cannot take for granted that RNAPII transcription from Pneumocystis is similar to other fungal species, since the transcriptional mechanisms are completely unknown yet and Pneumocystis is an obligated mammalian pathogen.

  • Specific comments. Thank you very much for the question. Those studies on the two forms of TFIIB, which are open (active) and closed (inactive) have been done mainly using mammalian TFIIB, and we do not know whether SpTFIIB or PcTFIIB can adopt those structures, since there are not studies on that issue. However, we know that the chimeric acidic activator Gal4-VP16 can activate in vitro transcription in a fission yeast whole cell extract. A few transcriptional activators have been described in Pneumocystis, but the mechanisms by which they work have not been described further. We could speculate, based on the structural similarity of SpTFIIB and PcTFIIB with human TFIIB, that a similar mechanism could operate and PcTFIIB can adopt an open (active) structure in the presence of fission yeast transcriptional activators, since it can partially function with the fission yeast RNAPII transcription machinery.
  • Figure 1: We have changed the color of the domains in Figure 1A according reviewer 1 suggestions. 3D structure of cerevisiae TFIIB was added to the new version of the figure.
  • Figure 2: The PCR figure has been included. Regarding to the internal control in the Western Blots, we added a housekeeping gene product such as tubulin. However, we previously quantified the amount of total protein by the Bradford assay to load equal amounts of total proteins. The colony growth on agar plates was included and we have changed the text from “small colonies” to “slow growth in YPD-agar” that is more accurate to our observations. In addition, we have added a panel showing the quantification of the length of the cells comparing wild type and mutant cells. Thank you very much for the suggestions.
  • Figure 3: We purified both recombinant TFIIB from coli. We purified recombinant TFIIB of Pneumocystis and fission yeast. Panel A shows the purification of PcTIIB from E. coli and Panel B shows an SDS-PAGE (Coomassie Blue stained) of PcTFIIB (from Panel A) and purified recombinant SpTFIIB. Panel C shows the Western blot analysis of both TFIIBs, which react with anti-His monoclonal antibodies, since both recombinant proteins possess a His-Tag to facilitate purification on Ni-NTA-agarose columns. Purity of the recombinant TFIIBs were based on gel of Panel A and Panel B. Line 211 was changed according suggestion. In panel 3D, IIF 1X corresponds to 20 ng of recombinant SpTFIIF, while IIF 2X corresponds to 40 ng of recombinant SpTFIIF.

Regarding the mobility of the TBP-TFIIB-RNAPII/IIF, we should expect that result in a sizeable shift. However, we should not forget that the mobility of a protein-DNA complex depends also on size of the DNA probe, protein charge, bending of the DNA and concentration of the polyacrylamide gel used to separate the complexes. In Panel D, we can see clearly that the addition of RNAPII/TFIIF results in a qualitative and quantitative different shifted complex. It is true that doubts could be casted whether RNAPII is into the complex, however those experiments were done with highly pure fission yeast RNAPII and purified recombinant GTFs, therefore, the possibility of an artefact is ruled out. Anyway, in the original experiment (same assay and film) we included that control, using the monoclonal antibody SWG16 (anti-CTD), which is able to inhibit the formation of the TBP-TFIIB-RNAPII/TFIIF-Promoter complex. That control is now shown in the figure. Also, in the original experiment we included a control omitting TFIIB, and it can be observed that TBP-RNAPII/TFIIF does not form any visible complex. Neither TBP-PcTFIIB-RNAPII-TFIIE can form stable complexes on the promoter. Finally, it is not worrisome at all that we used 1-3 ng of labelled probe, since that is the concentration used in most of the EMSA used to detect complexes on the TATA box using GTFs and RNAPII. Also, the amount of recombinant GTFs is in between 5-40 ng and 100-200 ng of RNAPII.

  • Figures 4-6: Figures were modified according to the reviewer comments. Than you very much for your suggestion and corrections.

Reviewer 3 Report

The manuscript by Rojas et al. reports that TFIIB from Pneumocystis carinii can replace the orthologous transcription factor in Schizosaccharomyces pombe in vivo and in in vitro transcription assays.

In addition to in vitro transcription assays, the authors also performed EMSA and primer extension experiments to investigate more in detail the mechanistic properties displayed by P. carinii TFIIB in this heterologous system. In my opinion, neither of these two experiments allows to draw firm conclusions, because of the lack of appropriate controls and/or the use of  inadequate experimental conditions.

More specifically:

1 (1)  In the EMSA experiments reported in Figure 3D, control lanes containing RNAPII/TFIIF (+/-TFIIE) in the absence of TFIIB are lacking. In the absence of such controls, it cannot be concluded that the supershifted complex observed in the last four lanes is due to TFIIB-mediated recruitment of Pol II /TFIIF. It should not be taken for granted that in this system RNAPII/IIF is not able to produce retarded complexes even in the absence of TFIIB.

2 (2) The primer extension experiment has not been carried out properly. Primer extension analysis should be carried out under conditions allowing for mapping the transcript 5’ end at single nucleotide resolution, using for example sequencing reactions carried out with the same labeled primer and run in parallel (as it is reported in two papers cited by the authors –refs. 20 and 31). In the absence of such conditions, it can not be concluded that the in vivo TSS observed in the replaced strain is the same as in wt strain. Moreover, from what can be seen in the gel in figure 6A, in the case of adh it seems that the extension products in wt and replaced strain have slightly different mobilities.

Author Response

Reply to the reviewers:

First, we would like to thank both reviewers for the careful and critical revision of our manuscript. Those comments will certainly improve the contents of the submitted manuscript.  Below you can find the answers to each comment, and we hope that now the manuscript would be ready for publication.

Reviewer 3

Thank you very much for the observations made. You can find our reply below.

  1. We do not completely agree with the statement that we cannot draw conclusions from the EMSA and primer extension experiments. We have included the appropriate controls (also asked by reviewer 2) to demonstrate that in the absence of PcTFIIB complexes are not formed on the TATA promoter, moreover, the shifted complexes contain RNAPII, since an antibody (SWG16 anti-CTD) can block formation of TBP-TFIIB-RNAPII/TFIIF-Promoter complex. Those experiments were done in the original one and it is shown in Panel D of Figure 3. Also, TFIIE it is not recruited to the complex in the absence of TFIIF as it is shown in the figure. That control was also asked by reviewer 2 and it is included now. It should be remembered that all of the GTFs are recombinant and RNAPII is highly pure, although traces of TFIIF might copurify, due to the tight association of TFIIF with RNAPII.
  2. The primer extension is an issue. As you might know by now, there are not suppliers of Sanger sequence kits, either by using PCR or Klenow fragment. Everybody has shifted to automatic sequence, since is cheaper and less troublesome than manual Sanger sequence. For routine sequence we use outside international facilities. Moreover, the cost and availability of radiolabeled nucleotides is a big problem in Latin America laboratories, or at least in Chile. Therefore, we could not perform any manual sequence to run together with the primer extension products. In any case, the fission yeast adh gene has three main TSS in vivo, separated by 5 - 8 nucleotides and those are clearly seen in the figure. Moreover, and most important, there are quite big changes in the TSS if we compare the wild type strain and the replaced one. Also, there is not big changes in the TSS usage comparing both strains. Additionally, neither reviewer 1 nor reviewer 2 have raised any issues on that experiment.

Reviewer 4 Report

The manuscript by Rojas et al., describes TFIIB of P. Carinii and demonstrates that P. Carinii TFIIB can functionally replace S. pombe TFIIB both in vitro and in vivo. P. Carinii TFIIB exhibits structural similarity to S. pombe TFIIB; can complement S. pombe TFIIB in vivo; and can substitute for S. pombe TFIIB in an in vitro transcription reaction. S. pombe cells expressing P. Carinii TFIIB, however, exhibit reduced growth rate and altered cell morphology. These results make an interesting scientific story. Authors, however, need to take care of the following minor issues:

(1)  Lines 149-150 states that….” The parental control strain does not show any reactivity to the anti-His antibody (Figure 2B)”… There is, however, no lane in Fig 2B for parental strain. All six lanes are for transformant and show reactivity with anti-His tag antibody.

(2)  In lines 156 and 159, figure citation is wrong. There is no Fig 2F. Please correct labelling.

(3)  Lines 156-157……”…colonies on YPD-agar plates are smaller than the wild type strain (data not shown)…..” For benefit of readers please show the data that there is a difference in size of cells between wild type and transformants when grown on YPD plates.

(4)  Lines 192-200 describes EMSA results shown in Fig. 3D. Please include lane numbers when citing Fig 3D so that it is easier for readers to understand results.

(5)  In Fig. 4, identity of antibodies used in Western blot must be indicated.

(6)  The sections of discussion are redundant with results. Please remove these redundancies.  

Author Response

Reply to the reviewers:

First, we would like to thank both reviewers for the careful and critical revision of our manuscript. Those comments will certainly improve the contents of the submitted manuscript.  Below you can find the answers to each comment, and we hope that now the manuscript would be ready for publication.

Reviewer 4:

Thank you very much for the observations made. You can find our reply below.

  1. We have indicated in the legend of the Figure 2 that “-“ symbol in lane 1 of the gel of panel 2C (new version of the Figure 2) corresponds to the analysis of protein extracts from wild type cells, that show only reaction to anti-tubulin antibody.
  2. We have corrected the wrong citation. Thank you very much for the correction.
  3. We have corrected the text indicating that the slow growth of mutant cells are evidenced in liquid and solid media. We have added a new panel in figure 2 showing a dot-plate experiment to complement the text description.
  4. We have indicated in the text the reference to the lanes of the EMSA showed in Figure 3. Thank you for your suggestion.
  5. We have indicated in the legend of the figure 4 the antibodies used in the experiments.
  6. We have removed in Discussion section the text related to results. Thank you very much for the suggestion.

Round 2

Reviewer 3 Report

I understand the difficulties in performing experiments. Nonetheless, the data of the primer extension experiment remain unconvincing. The different bands are not sufficiently resolved, and in the case of ADH1 the extended products in the wt appear to have slightly lower mobility that the corresponding band in the mutant. (And there isn't any scheme of the TSS region of the two genes helping to interpret the primer extension profile). Based on this experiment, it can not be concluded that replacement of SpIIB by PcIIB does not change the in vivo TSS. I am not convinced that a satisfactory primer extension experiment can no more be performed. ddNTPs are still being sold, and there are very recent publications showing primer extension experiments done properly (see here for an example: https://doi.org/10.1016/j.micpath.2022.105460). 

Author Response

Dear Reviewer:

Thanks for your comments. Here we present the answer to your comments.

The TSS from the adh1 gene promoter has been determined by Russell and by Gralla and coworkers (10.1016/S0022-2836(02)00329-7). Gralla and coworkers have determined three TSS (Figure 5C) and it can be seen a major start at +1 (A), and two minor at – 10 (G) and at + 4 (A). Our results also show three major start sites, and in the wild type the major start at +1 and two additional TSS located downstream and upstream from +1. The major start site in the mutant strain is only slightly offset from the major start of the wild type and presumable transcription starts at +2 in the mutant strain. In any case, transcription in the replaced strain starts in the same window as the wild type and more important within a narrow window (20-40 bp) downstream from the edge of the TATA box and this would not change the mRNA expression nor the translation of the mRNA.

On the other hand, the nmt1 gene promoter has two start sites, one at +1 (A) and minor one located downstream from +1. We see only two start sites from the nmt1 gene promoter and those do not change in the replaced strain. According to this analysis we have toned down our conclusions and changed the text of the section 2.6. We have added a scheme with the sequence and the known start sites that have been mapped in adh1 and nmt1 promoters to the Figure 6.

We have mentioned earlier the difficulties to perform this experiment mainly due to three reasons:

  1. The excessive cost and difficulties to get on time any radioisotopes. A vial of radioisotope costs USD 6,000 in Chile.
  2. The lack of any available commercial sequencing kit in Chile. It is true that still ddNTPs are sold, but we have to determinate the right concentration to perform a manual Sanger sequence and we do not have the experience, since we have always worked with kits to carry out reactions for manual Sanger sequence.
  3. We have only ten days to reply and upload the manuscript.

Round 3

Reviewer 3 Report

The authors have made an effort to address my further criticism, both by introducing a clarifying scheme in Figure 6 and by providing additional reasonable arguments.